# A new method for the study of biophysical and morphological parameters in 3D cell cultures: Evaluation in LoVo spheroids treated with crizotinib

Azzurra Sargenti[1], Francesco Musmeci[1], Carola Cavallo[2], Martina Mazzeschi[3], Simone Bonetti[1], Simone Pasqua[1], Francesco Bacchi[1], Giuseppe Filardo[4], Daniele Gazzola[1], Mattia Lauriola[3], Spartaco Santi[5,6]*

**1** Cell Dynamics iSRL, Bologna, Italy, **2** RAMSES Laboratory, IRCCS Istituto Ortopedico Rizzoli, Bologna, Italy, **3** Department of Experimental, Diagnostic and Specialty Medicine (DIMES), University of Bologna, Bologna, Italy, **4** Applied and Translational Research Center, IRCCS Istituto Ortopedico Rizzoli, Bologna, Italy, **5** Institute of Molecular Genetics "Luigi Luca Cavalli-Sforza", Unit of Bologna, CNR, Bologna, Italy, **6** IRCCS Istituto Ortopedico Rizzoli, Bologna, Italy

☯ These authors contributed equally to this work.
* spartaco.santi@cnt.it

**Data Availability Statement:** All relevant data are within the manuscript and its Supporting Information files.

## Abstract

Three-dimensional (3D) culture systems like tumor spheroids represent useful *in vitro* models for drug screening and more broadly for cancer biology research, but the generation of uniform populations of spheroids remains challenging. The possibility to properly characterize spheroid properties would increase the reliability of these models. To address this issue different analysis were combined: i) a new device and relative analytical method for the accurate, simultaneous, and rapid measurement of mass density, weight, and size of spheroids, ii) confocal imaging, and iii) protein quantification, in a clinically relevant 3D model. The LoVo colon cancer cell line forming spheroids, treated with crizotinib (CZB) an ATP-competitive small-molecule inhibitor of the receptor tyrosine kinases, was employed to study and assess the correlation between biophysical and morphological parameters in both live and fixed cells. The new fluidic-based measurements allowed a robust phenotypical characterization of the spheroids structure, offering insights on the spheroids bulk and an accurate measurement of the tumor density. This analysis helps overcome the technical limits of the imaging that hardly penetrates the thickness of 3D structures. Accordingly, we were able to document that CZB treatment has an impact on mass density, which represents a key marker characterizing cancer cell treatment. Spheroid culture is the ultimate technology in drug discovery and the adoption of such precise measurement of the tumor characteristics can represent a key step forward for the accurate testing of treatment's potential in 3D i*n vitro* models.

**Funding:** This work has been partially supported by grants by the Ministry of Economic Development-AGRIFOOD PON I&C 204-2020"Development of a technological platform for the functional testing of nutraceutical molecules". Project Nr F/200110/01-03/X45-CUP B61B19000580008 to Cell Dynamics iSRL. The funder provided support in the form of part of salaries for authors AS, FM, SB, SP and DG, but did not have any additional role in the study design, data collection and analysis, decision to publish, or preparation of the manuscript. The specific roles of these authors are articulated in the 'author contributions' section.

**Competing interests:** The authors of Affiliation 1 (AS, FM, SB, SP, FB and DG) are employed by Cell Dynamics iSRL company. This does not alter our adherence to PLOS ONE policies on sharing data and materials. The remaining authors (CC, MM, GF, ML and SS) declare that the research was conducted in the absence of any commercial or financial relationships that could be construed as a potential conflict of interest. The authors declare that a Patent Application (No. 102020000006031) incorporating parts of this work has been filed. This does not alter our adherence to PLOS ONE policies on sharing data and materials. Daniele Gazzola, Simone Bonetti, Domenico Andrea Cristaldi, Azzurra Sargenti and Francesco Musmeci are the inventors of patent No. 102020000006031.

## Introduction

Three-dimensional (3D) culture systems are widely recognized as an *in vitro* model with a close resemblance to solid tumors *in vivo*. Tumor spheroids derive from the self-organization of cancer cells into 3D tumor-like spherical structures [1]; such tissue architecture confers significant properties with respect to the monolayer disposition, which are crucial in terms of tumor cell fate modulation [2,3]. In fact, 3D models recapitulate the complex cellular organization of the tumor bulk. Both cell–cell adhesions [4] and cell–matrix interactions, along with a *de novo* synthesis of extracellular matrix (ECM) proteins [5,6], contribute to the generation of nutrient and signal gradients typical of solid tumors, thus offering a clinically relevant study model [7–11]. For these reasons, spheroids represent a useful tool for drug screening [12,13] and more broadly for cancer biology research [5,14]. Compared to 2D monolayer cells, spheroids display a 3D morphology, which is closer than 2D monolayer cell system to the tumor bulk *in vivo*. Spheroids are useful for toxicological studies, as concerning drug penetration and absorption, thus providing insights on drug diffusion [15] and exhibit better immuno-modulatory, proliferation, and activation abilities than 2D cultures [16].

Despite the great potential of these 3D models, the generation of uniform populations of spheroids in terms of size and shape remains challenging. In fact, cellular density of the spheroids coupled with the size-dependent mass transport are critical factors affecting cell functions, as they influence drug penetration and their overall physiology [5,17]. Thus, the possibility to study spheroid properties by means of a precise characterization would allow a better understanding of the results with respect to their features, in the end increasing the reliability of these models. Research efforts are needed to provide a more appropriate and reliable data analysis and interpretation [3,18–20]. In particular, the dynamic changes of spheroids over time and under treatment are usually monitored by the software elaboration [21,22] of bi-dimensional images, by measuring parameters such as area, radius ratio, roundness, and density [3,20,23]. This approach is not applicable to the complexity of spheroids with an irregular morphology and density, and the 2D projection might alter the actual data related to the 3D structure.

One of the main limitations for a comprehensive characterization of 3D models is related to the mass density and its correlation to the volume. To address this issue, we recently developed a new device (W8, CellDynamics iSRL) and relative analytical method, for the accurate, simultaneous, and rapid measurement of mass density, weight, and size of spheroids generated from different cell types [24,25]. In general, the device is able to measure all types of spheroids derived from cancer and primary cells which meet the operative range between 50 and 500 μm in diameter. Furthermore, characterization of large cellular aggregates morphology is critically affected by the cells density that may alter the penetration of light throughout the z-depth, with consequent light scattering and image distortion [17]. To bypass this limitation, we performed a tissue clearing process that removes lipid substances, combined with a confocal microscopy analysis with a refractive index matching that reduces spherical aberration.

The innovative use of the new fluidic-based measurements combined with confocal analysis and protein quantification allow to investigate 3D models in detail. A robust morphological characterization supported by biophysical parameters correlation, tested in a challenging and clinically relevant applicative scenario, allowed to unveil a previously unappreciated feature in the tumor spheroids.

In this study, the LoVo colon cancer cell line forming spheroids was treated with crizotinib (CZB) an ATP-competitive small-molecule inhibitor of the receptor tyrosine kinases. This inhibitor, already been proved to impact LoVo spheroids morphology [26,27], was employed

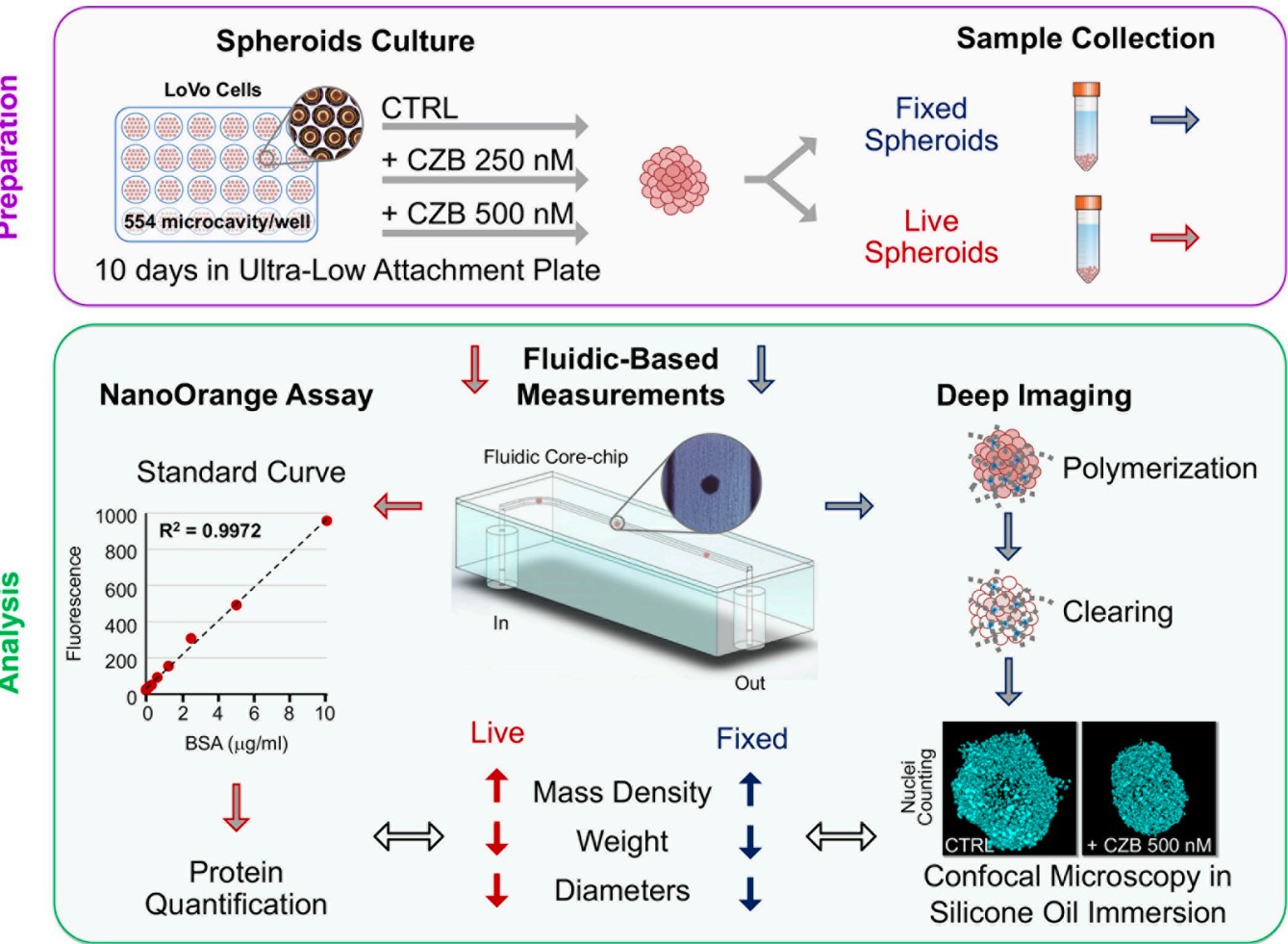

**Fig 1. Scheme of the research strategy.** Experimental outline of the research work, from spheroid culture and treatment with CZB to setting up the analyses on fixed and live samples. The LoVo cells were cultured in Ultra-Low Attachment Microcavity Plate in presence of 250 and 500 nM of CZB. At 10 days fully mature and organized spheroids were harvested, for the analysis procedures. Live and fixed samples were evaluated with a new fluidic device (W8) for the accurate, simultaneous, and rapid measurement of mass density, weight, and size. These parameters were correlated with protein quantification (NanoOrange Assay) and deep imaging analysis performed on clarified samples.

to assess the correlation between biophysical and morphological parameters in both live and fixed cells (Fig 1).

## Materials and methods

### Device and analytical method for spheroid characterization

Recently, we standardized a new device (W8, CellDynamics iSRL), and relative analytical method, for the accurate, simultaneous, and rapid measurement of mass density, weight, and size of cellular spheroids. The technique is based on the detection of the terminal velocity, which is derived by the tracking the sample's motion when free-falling into a vertical flow-channel while the flow is at rest (see S1 Video). This is achieved by using a customized software that relies on a specific physical method, combined with detailed statistical analyses [24]. Briefly, the device is composed of a fluidic-core chip, equipped with a bright field imaging setup, a peristaltic pump, a temperature sensor and a flow-circuit for the introduction and

elimination of samples. Bright field images are acquired to measure the radius of the sample and to track its motion. This physical method detects mass density and simultaneously weight of for each spheroid. Raw data can be found in S1 Dataset. The potential of these fluidic-based measurements was investigated in a clinically relevant model of LoVo colon cancer cell line forming spheroids, to characterize mass density, weight and diameter of the spheroids, and their correlation with the spheroid biophysical parameters as confirmed by protein quantification and deep imaging investigation after sample optical clarification.

## Spheroids culture method

LoVo, a cell line derived from a human colon adenocarcinoma was cultured in DMEM (Sigma-Aldrich, St. Louis, MO) with 10% FCS (Euroclone, Milan, Italy) supplemented with 1% penicillin/streptomycin (Thermo Fisher Scientific, Waltham, Massachusetts, USA). Cells were expanded and maintained as a monolayer at 37°C and in atmosphere containing 5% $CO_2$. LoVo cells were cultured in a 24-well Black/Clear Round Bottom Ultra-Low Attachment Microcavity Plate (Corning® Elplasia®Plates, Corning Inc., NY, USA) according to the manufacturers' instructions. Microcavities allow cells to self-assembly into large numbers of uniform, size-controlled 3D spheroids depending on microcavity diameters and geometry [13,28]. Cells were seeded at the concentration of $1x10^6$/wells and cultured for 10 days in DMEM (Sigma-Aldrich, St. Louis, MO) with 10% FCS in presence of 250 and 500 nM of CZB (MedChem Express), at 37°C and 5% $CO_2$. The inhibitor was added to the medium simultaneously with the seeding as previously described [27]. LoVo cells grown in DMEM (Sigma-Aldrich) with 10% FCS were used as control (CTRL). At 10 days fully mature and organized spheroids were harvested, for the analysis procedures.

## Measurement of spheroid biophysical parameters

For the analyses, live and fixed LoVo spheroids were used. At 10 days spheroids were harvested, centrifuged, and washed twice in Dulbecco's phosphate-buffered saline (DPBS) 1X w/o $Ca^{2+}$ & $Mg^{2+}$ (Corning® Life Sciences). Fixed samples were exposed to PFA 4% at room temperature (RT) for 3 hours and then washed twice with Dulbecco's phosphate-buffered saline (DPBS), 1X w/o $Ca^{2+}$ & $Mg^{2+}$ (Corning® Life Sciences). Before the measurements, all the samples were resuspended in 3.5 mL of DPBS at low concentration (<200 spheroids/mL), transferred in a centrifuge conical tube and analyzed according to the previous protocol for biological samples [24]. A minimum of 10 single spheroids were analyzed for every tested condition and values were extrapolated from at least 10 repetitions.

## Protein quantification

At 10 days spheroids were harvested, washed twice in Phosphate Buffered Saline (PBS) and centrifuged at 6000 rpm for 5 minutes. LoVo spheroids were lysed in a buffer containing 20 mM TRIS–HCl (pH = 7.5), 1% SDS, 1 mM $Na_3VO_4$, 1 mM PMSF, 5% beta-mercaptoethanol and protease inhibitors. After sonication and centrifugation, proteins were quantified by NanoOrange (Thermo Fisher Scientific) methods according to manufacturer's instructions. Briefly, samples were diluted in the NanoOrange working solution and incubated at 90°C for 10 minutes, protected from light. Successively, samples were cooled at room temperature for at least 20 minutes and transferred in a 96 wells plate. Fluorescence was read at 470ex–570em nm wavelength, using a Spectra Max Gemini plate fluorometer (Molecular Devices, Sunnyvale, CA). Protein concentration was determined using the reference standard curve ($R^2$, coefficient of determination was reported in Fig 1).

## Spheroids clearing

To obtain an optically transparent tissue, we employed the X-Clarity™ Tissue Clearing System (Logos Biosystems, Inc. South Korea), which is able to extract lipids through electrophoresis creating a tissue-hydrogel hybrid that permits light penetration and multiple staining [29]. Spheroids were fixed in PFA 4% for 3 hours at RT, rinsed three times with PBS and stored at 4˚C. Successively, samples were completely immersed in the X-CLARITY Hydrogel-Initiator solution (Logos Biosystems, Inc. Anyang-Si, Gyunggi-Do, South Korea) for at least 18 hours at 4˚C, according to manufacturer's instructions. Samples were polymerized with the X-Clarity™ Polymerization System (Logos Biosystems) for 3 hours with a vacuum of 90 kPa and a temperature of 37˚C. After polymerization, spheroids were gently shaken for 1 minute, rinsed several times with PBS and stored at 4˚C. Subsequently, samples-hydrogel hybrids were immersed in an Electrophoretic Tissue Clearing Solution (Logos Biosystems) and placed for 4 hours in the X-Clarity™ Tissue Clearing System (Logos Biosystems) with the following settings: current 0.8 A, temperature 37˚C, pump speed 30 rpm.

After PBS washing, the samples were then placed on Fluorodish Cell Culture Dishes (WPI), with optical quality glass bottom and directly stained with 50 μg/ml Phalloidin-TRITC (Sigma-Aldrich, P1951) and 0.1μg/ml 4',6-Diamidino-2-phenylindole (DAPI, Sigma-Aldrich, D9542). Spheroids were then mounted with a mixture of X-CLARITY mounting solution (Logos Biosystems) and 1,4-Diazabicyclooctane (DABCO) (Sigma-Aldrich).

## Deep imaging and analysis

Imaging was performed on clarified samples with a Nikon A1-R confocal microscope, equipped with a 25× (silicone immersion, 1.05 NA) objective and with 405 and 561 nm laser lines to excite DAPI (1:500, #D9542, Sigma-Aldrich) and Alexa-568 Phalloidin (1:100, #A12380, Thermo Fisher Scientific) incubated for 30 minutes at room temperature. The silicone immersion objective provides a long working distances (0.8 mm) to perform deep imaging and reduce spherical aberration which arises from refractive index mismatches occurring along the optical path between immersion media and mounting solution: in this case n = 1.40 and n = 1.46, respectively. Z-stacks were collected at optical resolution of 210 nm/pixel, stored at 12-bit with 4096 different gray levels, pinhole diameter set to 1 Airy unit and z-step size to 1 μm. The data acquisition parameters were setup in a fixed manner, such as laser power, gain in amplifier and offset level. Confocal images were processed using the Richardson-Lucy deconvolution algorithm. Nuclei counting was realized using Nikon NIS Elements AR software (GA3 analysis module). The images were cleaned from noise and subsequently aligned along the Z axis. The nucleus centres were identified with the function BrightSpot detection 3D. The recognized centres were then grown to the margin of the nuclei in order to identify the stained region (GrowRegions) then smoothed with a radius parameter of 0,332 μm (Smooth) and a SeparateObject with a value of 2 counts. The area identified where then used for counting the Total Nuclei (ObjectCount). A minimum of 10 single spheroids were analyzed for every tested condition and performed in triplicate.

## Statistical analysis

As previously reported [24], on the obtained mass density, weight and diameter outputs, first the Shapiro–Wilk test was performed to analyze the distribution of the dataset based on skewness and/or kurtosis. For all the cases that resulted in a non-normal distribution, descriptive statistics box plots (Tukey method plots) were then carried out for determining and eliminating outliers' values (K > 1.5). After outlier's elimination, the Shapiro–Wilk approach was reiterated to confirm the normal distribution. Data are presented as mean ± SD. Statistical

analysis was performed using two-tailed unpaired Student's *t*-test. The cut-off value of significance is indicated in each figure legend.

To evaluate the fixation effect on mass density, weight and diameter, we used the "r" Pearson correlation coefficient (range value from -1 to 1). We analyzed the average value of mass density, weight and diameter for every drug concentration. We estimated the total effect of fixation on W8 output data with one-way ANOVA test as indicated in figure legend.

## Results and discussion

### Device characterization of live and fixed spheroids biophysical parameters

To evaluate the potential of the new approach, the device and analytical models were applied to a clinically relevant *in vitro* model, LoVo spheroids, where the effects of CZB treatment on the biophysical parameters were studied. The same characteristics were tested and compared both in live and fixed samples conditions, in order to exclude any technical problem with the fixation which could impair the biophysical parameters analyzed.

We found that in live samples CZB treatment significantly decreases weight and size (Fig 2B and 2C) at concentrations of 250 and 500 nM, compared to CTRL. This is in line with an antiproliferative activity of the inhibitor [27]. On the other hand, under the same treatments, mass density displays an opposing result, with a significant increase in the density of the spheroids' bulk (Fig 2A).

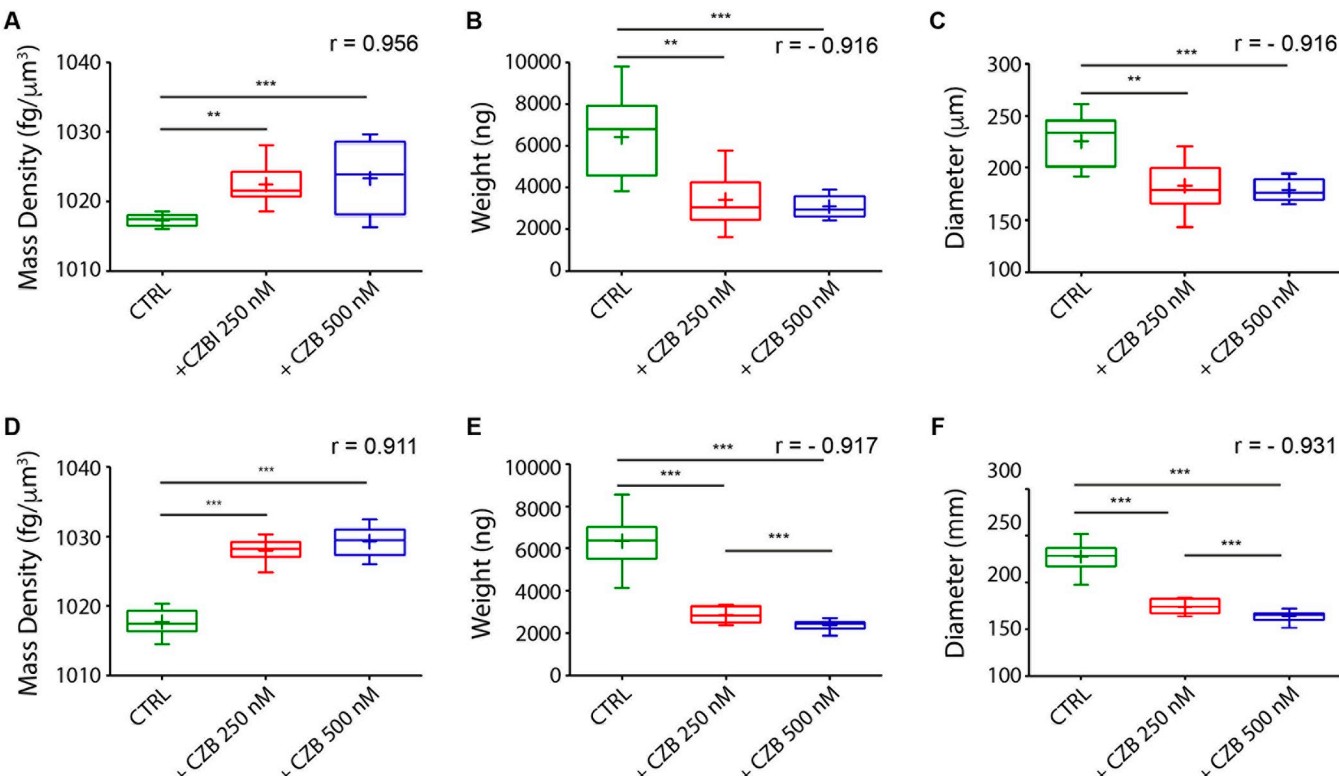

**Fig 2. Measurement of mass density, weight, and diameter of LoVo spheroids treated with CZB.** Measurements of mass density (A and D), weight (B and E), and diameter (C and F) of live (top panels) and fixed (bottom panels) LoVo spheroids after 10 days of treatment with CZB 250 nM (shown in red) and 500 nM (shown in blue) and relative controls (shown in green). Data are graphically depicted in box-and-whisker plots and the lines, extending from the boxes, indicate variability outside the upper and lower quartiles. **p< 0.01 and ***p< 0.001. The "r" Pearson correlation coefficients, able to measure the degree of relationship between the value of mass density, weight and diameter and the different drug concentrations, are indicated for each graph. The F-ratio value obtained from one-way ANOVA test estimating the total effect of fixation on W8 output data, is 0.00027, the result is not significant at p < 0.01.

We also tested the effect of the fixation protocol (Fig 2D–2F). The results agree with the living imaging, as we observed a significant reduction of weight and size of treated samples compared to CTRL (Fig 2E and 2F), and a marked enhancement of mass density (Fig 2D).

These findings confirm that PFA fixation protocols did not altered the parameters under investigation, with respect to MRI parameters [30]. This information is relevant for the feasibility and usability of the assay, allowing to test samples stored in different laboratories and in multiple experiments over-time. Finally, the assay was able to clearly show an effect of CZB on the cellular density of the spheroids, which is usually underestimated in the spheroid measurements. We supposed that the treatment may impact on the tumor mass absorption and drug metabolism *in vivo* by increasing cell-cell adhesion in the core of the spheroids. The new evaluation method allows to characterize spheroid properties and thus to derive more reliable conclusions to translate the findings of this *in vitro* model into the clinical setting.

## Protein quantification

Protein quantification performed by NanoOrange assay was used to further characterize spheroids and investigate if these findings correlate with those provided by the new proposed spheroid characterization model. Protein quantification confirmed an overall decrease in protein levels in the samples treated with CZB, compared to the untreated conditions (Fig 3A).

These data suggest that spheroid size is correlated to the amount of protein, which most probably is an indirect measurement of the number of cells composing the sphere, thus supporting the results of the spheroid characterization in terms of mass density, weight, and size provided by the W8 fluidic system.

## Deep imaging and analysis

Bright field images from clarified samples (Fig 3B upper panel) show a decrease in size and an increase in optical density (as measured in Fig 3C), indicating a progressive growth in cell-cell compactness. By confocal microscopy, we analysed the 3D cell architecture of LoVo spheroids. To assess the loss of fluorescence and resolution associated with standard confocal imaging of spheroids throughout the z-depth, we imaged cleared spheroids combined with a silicone immersion oil objective suitable for high-resolution imaging. The elevated penetration depth of cleared samples and the reduced spherical aberration obtained by means of the silicone objective [31] permitted to study the relationship between the spheroid density and the morphology of LoVo spheroids' bulk. After 10 days of cells growth in culture, the untreated spheroids appeared frayed and with low nuclear density on the outermost layer, while the ones under CZB treatment (250 and 500 nM) displayed a progressive increase of 3D nuclear density and cell-cell compactness (Fig 3B middle panel). Fluorescence phalloidin staining reveal the expression levels of actin cytoskeletal proteins is low and is not affected by CZB treatment. The quantitative analysis of imaged spheroids, after an accurate automatic segmentation of DAPI signal, indicates a significant decrease in the total number of cell nuclei (Fig 3D) and in spheroid volume (Fig 3E) of CZB treated samples compared to CTRL. Conversely, under the same treatments, cell nuclei density (the relationship between the number of nuclei and the spheroid volume) displays an opposing result, with a significant increase in the density of the spheroids' bulk (Fig 3F). Altogether, these results are in line with the previously characterized spheroid properties, confirming the potential of the new proposed analyses to provide complementary information for a reliable investigation and interpretation of the results obtained in 3D *in vitro* models.

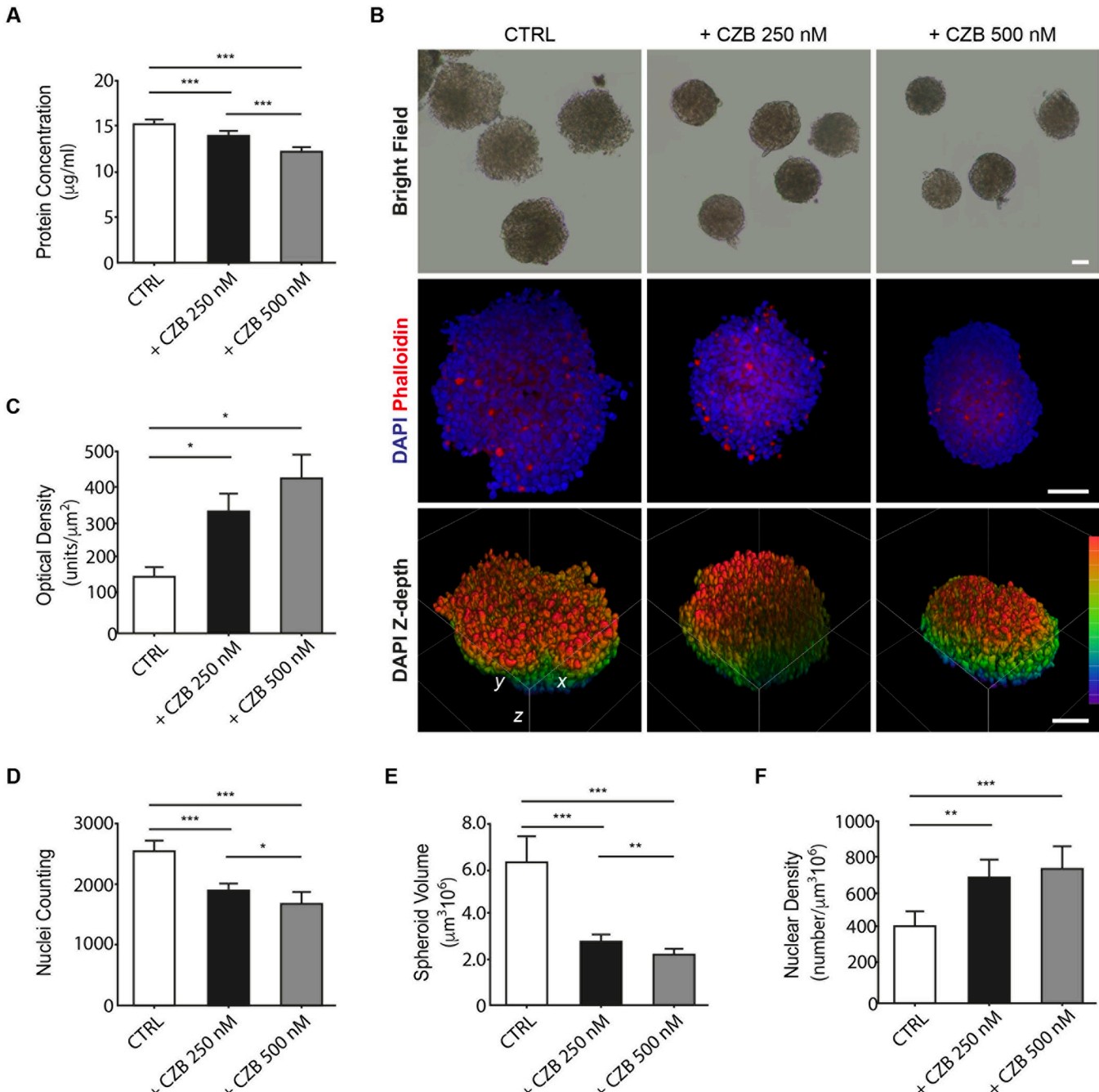

**Fig 3. Protein quantification and imaging analysis of CZB treated LoVo spheroids.** A) Protein quantification performed by NanoOrange assay of CZB 250 nM and 500 nM, and relative control. Data are presented as mean± SD. Statistical analysis was performed using two-tailed unpaired Student's *t*-test. *p< 0.05, **p< 0.01, and ***p< 0.001. B) Representative images of LoVo spheroids exposure to different doses of CZB after clearing procedures. The upper panel shows bright field imaging from control and treated spheroids at low magnification. The middle panel exhibits confocal imaging of DAPI (in blue) and phalloidin (in red) staining for nuclei and actin respectively. The lower panel provides volume view of DAPI signal in 3D z-depth confocal rendering reported as a scale of colors (palette on the right). Scale bars 50 μm. C) Optical density of fixed LoVo spheroids, calculated as ratio between the mode of absorbance (subtracted to the background) and the spheroid area after 10 days of treatment with CZB 250 nM (histogram in black), 500 nM (histogram in gray) and relative control (histogram in white). Statistical analysis was performed using two-tailed unpaired Student's t-test. *p< 0.05, **p< 0.01, and ***p< 0.001. D) Total cell nuclei counting of fixed LoVo spheroids after 10 days of treatment with CZB 250 nM (histogram in black), 500 nM (histogram in gray) and relative control (histogram in white). Statistical analysis was performed using two-tailed unpaired Student's *t*-test. *p< 0.05, **p< 0.01, and ***p< 0.001. E) Volume of fixed LoVo spheroids after 10 days of treatment with CZB 250 nM (histogram in black), 500 nM (histogram in gray) and relative control (histogram in white). Statistical analysis was performed using two-tailed unpaired Student's *t*-test. *p< 0.05, **p< 0.01, and ***p< 0.001. F) Volumetric cell nuclei density of fixed LoVo spheroids, calculated as ratio between the total number of nuclei and the spheroid volume after 10 days of treatment with CZB 250 nM (histogram in

black), 500 nM (histogram in gray) and relative control (histogram in white). Statistical analysis was performed using two-tailed unpaired Student's *t*-test. *p< 0.05, **p< 0.01, and ***p< 0.001.

## Conclusions

Overall, these results support the potential of the new proposed methodology to investigate 3D models, with morphological characterization being supported by biophysical parameters correlation. This model allowed to unveil a previously unappreciated feature in the treated spheroids, which upon CZB inhibition display a strong increase in mass density, along with a reduction of size and weight. These results were further supported by the protein quantification and the deep imaging analysis that showed protein level decrease in line with a reduced proliferation, a decrease of the number of nuclei and the spheroid volume, but a strong increase in nuclear density and cell-cell compactness, according to mass density data. The new fluidic-based measurements allowed a robust phenotypical characterization of the spheroids structure, offering insights on the spheroids bulk and an accurate measurement of the tumor density. This analysis helps overcome the technical limits of the imaging that hardly penetrates the thickness of 3D structures. Accordingly, we were able to document that CZB treatment has an impact on the mass density, which represents an important marker characterizing cancer cell treatment. We speculate that the possible mechanism responsible for the increased spheroid mass density may be related to the enhanced cell-cell adhesion, as previously reported [27]. Indeed, we found that CZB treated cells displayed increased e-cadherin production, with consequent improved tight-junction organization. This might lead to an increase in the compactness of the spheroids. Furthermore, we cannot exclude that this phenotype is the consequence of the apoptotic effect registered with CZB inhibition, which might be responsible for a decreased spheroids size and increase in the bulk filling.

All these and further key characteristics in tumor development and response to treatment can be better investigated in 3D models. In this light, the results of the newly proposed methodology could serve as a stepping stone to further improve the 3D model field. The use of patient-derived 3D tumor models can be considered the next generation of in vitro models that can closely recapitulate patient's disease and can be used to screen drugs, to optimized the correct dosage, as well as to develop new personalized therapeutic strategies [32]. However, the disadvantages of 3D models include difficulties evaluating biophysical parameters and obtaining high-resolution imaging analysis due to hardly light penetration in the thickness of deep structures. While this study provided the proof of concept in spheroids to support this methodology, this promising approach should be also applied to more complex patient-derived 3D systems in the field of personalized medicine. Thus, future studies should aim at confirming the positive findings obtained in spheroids also in organoids and 3D models of increasing complexity, where the adoption of such precise measurement of the tumor characteristics could represent a key step forward for the accurate testing of treatment's potential in a clinically relevant model.

## Supporting information

**S1 Video. W8 measurement system analysis.** Representative video displaying the different phases of the W8 measurement system analysis: spheroid selection, stabilizing and centering phases, measurements process (2 repetitions), and the final sorting. The analysis channel is shown at low magnification (LQ camera, on the left) and at higher magnification (HQ camera, on the right). The frame rate speeded up about 3 times.
(MP4)

**S1 Dataset. W8 raw data.** Measurements raw data of mass density, weight, and diameter of live and fix LoVo spheroids treated with crizotinib 250 and 500 nM.
(PDF)

## Acknowledgments

The authors are grateful to Dr. Giacomo Cozzi (Nikon Italia S.r.l) and Dr. Emanuela Ariotti (Logos Biosystems Inc.) for skilful technical assistance.

## Author Contributions

**Conceptualization:** Carola Cavallo, Spartaco Santi.

**Data curation:** Azzurra Sargenti, Francesco Musmeci, Carola Cavallo, Francesco Bacchi.

**Formal analysis:** Francesco Musmeci, Martina Mazzeschi.

**Funding acquisition:** Simone Pasqua, Daniele Gazzola.

**Investigation:** Azzurra Sargenti, Carola Cavallo.

**Methodology:** Simone Bonetti.

**Project administration:** Spartaco Santi.

**Software:** Simone Bonetti, Daniele Gazzola.

**Supervision:** Simone Pasqua, Mattia Lauriola.

**Validation:** Azzurra Sargenti, Francesco Musmeci, Carola Cavallo, Mattia Lauriola.

**Visualization:** Francesco Bacchi.

**Writing – original draft:** Azzurra Sargenti, Giuseppe Filardo.

**Writing – review & editing:** Giuseppe Filardo, Mattia Lauriola, Spartaco Santi.

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
