## [Decision Letter · Decision Letter 0]

30 Mar 2021

PONE-D-21-04842

A new method for the study of biophysical and morphological parameters in 3D cell cultures: Evaluation in LoVo spheroids treated with TKI

PLOS ONE

Dear Dr. Santi,

Thank you for submitting your manuscript to PLOS ONE. After careful consideration, we feel that it has merit but does not fully meet PLOS ONE’s publication criteria as it currently stands. Therefore, we invite you to submit a revised version of the manuscript that addresses the points raised during the review process.

Please note that the PLOS data availability policy requires authors to make all data underlying the findings described in their manuscript fully available without restriction, with rare exception. The data should be provided as part of the manuscript or its supporting information, or deposited to a public repository.

I invite you to resubmit your manuscript after addressing the reviewers’ comments below and in a file attached in the submission system.  Please consider all issues mentioned in the reviewers' comments point-by-point and provide suitable rebuttals for any comments not addressed.

We look forward to receiving your revised manuscript.

Kind regards,

Irina V. Balalaeva, PhD

Academic Editor

PLOS ONE

Journal Requirements:

4. Thank you for stating the following in the Financial Disclosure section:

'This work has been partially supported by grants by the Ministry of Economic Development-AGRIFOOD PON I&C 204-2020“Development of a technological platform for the functional testing of nutraceutical molecules". Project Nr F/200110/01-03/X45-CUP B61B19000580008 to Cell Dynamics iSRL. '

We note that one or more of the authors have an affiliation to the commercial funders of this research study: Cell Dynamics iSRL

c. We also note that you have a patent relating to material pertinent to this article. In your amended statement of Competing Interests please declare this patent (with details including name and number), along with any other relevant declarations relating to employment, consultancy, patents, products in development or modified products etc. Please confirm that this does not alter your adherence to all PLOS ONE policies on sharing data and materials, as detailed online in our guide for authors http://journals.plos.org/plosone/s/competing-interests by including the following statement: "This does not alter our adherence to  PLOS ONE policies on sharing data and materials.” If there are restrictions on sharing of data and/or materials, please state these.

Please note that we cannot proceed with consideration of your article until this information has been declared.

Reviewers' comments:

Reviewer's Responses to Questions

**Comments to the Author**

1. Is the manuscript technically sound, and do the data support the conclusions?

Reviewer #1: Partly

Reviewer #2: Yes

Reviewer #3: Yes

Reviewer #4: Yes

2. Has the statistical analysis been performed appropriately and rigorously? 

Reviewer #1: Yes

Reviewer #2: Yes

Reviewer #3: Yes

Reviewer #4: Yes

3. Have the authors made all data underlying the findings in their manuscript fully available?

Reviewer #1: No

Reviewer #2: No

Reviewer #3: No

Reviewer #4: No

4. Is the manuscript presented in an intelligible fashion and written in standard English?

Reviewer #1: Yes

Reviewer #2: Yes

Reviewer #3: Yes

Reviewer #4: Yes

5. Review Comments to the Author

Reviewer #1: The manuscript is devoted to an actual issue of modern biomedicine, in particular the development of 3D cell tumor models and their implementation for drug testing. The authors provide some interesting data concerning LoVo spheroids and approaches to evaluate the efficacy of tyrosine kinases inhibitors in tumor spheroids. However, I have some concerns about this manuscript.

1. The novelty of this research should be clarified. The authors use classical approach to form spheroids (low adhesive surface) and commercial kits and equipment, except of fluidic-based device. However, the authors have described this device earlier (https://doi.org/10.3390/mi11050465). Therefore, I suppose that the real novelty of the research is the use of fluidic-based device for antitumor drug testing in tumor spheroids model.

2. The authors use a small ATP-competitive receptor tyrosine kinases inhibitor (TKI), but TKI is not a drug, but a class of drugs (doi:10.2174/138920009788897975). Please clarify which one has been used.

3. The choice of TKI concentrations is unclear, as well as incubation time (10 days?). If TKI was added to the cells before spheroids formation, it affected on single cell suspension, not on spheroids. Please clarify the protocol.

4. The use of conventional techniques (MTS, or XTT, or WST-1) could be useful to evaluate the viability of cells in spheroids after TKI treatment. Since these assays have been developed to measure cell viability, they are more sensitive then protein quantification, which is obviously the simple function of cell amount.

5. How many spheroids was in each well of 24-well plates? I suppose that several dozens, but in this case, Fig. 1 is confusing (one spheroid per well).

6. It is better to transfer Fig. 1 from Introduction to Results or even to Graphical abstract since it contains results.

7. Since the authors made a lot of effort to realize deep imaging, the results (three images of nuclei staining) seems insufficient. I propose to calculate the density of cell nuclei using the total area of spheroids to compare it with mass density data. The phalloidin staining is week and does not provide any information.

8. The authors discuss LoVo spheroids as a clinically relevant in vitro model, but in fact, spheroids are in vitro models only, while clinical significance is under discussion.

9 The authors discuss the problem of the generation of uniform populations of spheroids, but the manuscript does not provide any new approaches to generate uniform spheroids.

10. Since the authors found that fixation protocols did not alter the spheroids parameters under investigation, these data could be used as supplementary. The fixation protocol (4% PFA) is a standard one in cell biology due to minimal influence on cell properties, so the scientific value of these results are questionable.

In summary, I suppose that the manuscript could not be published in the current version. It contains interesting data concerning the increase of mass density in spheroids after treatment with TKI, but the scientific value is limited due to 1) insufficiency in experimental conditions (one drug in two random (?) concentrations in one time point), 2) absence of conventional techniques to compare, 3) lack of explanation of mass density increase effect. More experiments are required to demonstrate the importance of mass density as drug efficacy marker, and fluidic-based device as a sensitive approach to measure mass density in comparison to confocal or histology imaging and counting, as well as to conventional assays to evaluate the efficacy of antitumor drugs in 3D cultures (MTS or WST-1). The possible mechanism of increase of mass density in spheroids should also be discussed (cell shrinkage due to apoptosis?).

Reviewer #2: This research presented an alternative to an important difficulty in the evaluation of spheroids in three-dimensional form, as biophysical aspects mass density, weight, and size of spheroids and morphological aspects by confocal imaging and quantification for evaluation of total protein to correlate with cell mass decrease, reaching its objectives. It is a work of great impact for the area.

Material and methods

- about information from lines 103 to 106

“Cells were seeded at the concentration of 1x106 /wells and cultured for 10 days in DMEM (Sigma-Aldrich, St. Louis, MO) with 10% FCS in presence of 250 and 500 nM of TKI (MedChem Express), at 37 °C and 5% CO2. LoVo cells grown in DMEM (Sigma-Aldrich) with 10% FCS were used as control (CTRL). “

Was the spheroid produced simultaneously with the treatment? If not, clarify in the text - MATERIALs AND METHODS lines 103 to 106.

If produced simultaneously with treatment: I suggest evaluating the parameters, in spheroid produced without simultaneous treatment and treated after its formation.

The spheroid was produced for 10 days: It is known that the number of cells and the time of spheroid formation is different for each cell line. As the author mentioned about the difficulty to obtain spheroids of homogeneous size “ the generation of uniform populations of spheroids remains challenging”, I suggest to add in Material and methods about the analyzed characteristics of the spheroid to standardize the 10 days of spheroid formation.

RESULTS

About result of fig. 2: Measurements of mass density (A and D), weight (B and E), and diameter (C and F) of 2 methods : live (top panels) and fixed (bottom panels). Since they showed variation in values for each parameter for both methodologies, I suggest statistically evaluating this difference to be able to state "fixation protocols did not alter the parameters under investigation".

Reviewer #3: I enjoyed very much reading this work. The methods presented here add valuable elements to the toolbox for evaluating 3D microtumors (and microtissues in general). The technology introduced is very clever and elegant. The Figures are very well finished and the manuscript flows very nice.

I would like to recommend just a few things:

• The fluidic device is equipped with an brightfield setup. Could you add a panel in figure 1 or a supplementary video showing how the spheroids look like when passing through the channel?

• The results of protein concentrations with NanoOrange assay is repeated in Figure 1 and 3. Instead, present the results only in Figure 3, and use and schematic representation of the assay in Figure 1 (a microplate reader?).

• Figure 3. Could you present the nuclei count in histograms with error bars? Are these nuclei the total number in the spheroid (volumetrically)?

• Could you add an additional panel in Figure 3 or a supplementary video showing the volumetric nature of the sample?

• Please add titles to the axis (Y axis: Protein concentration (�g/ml); X axis: Sample)

• Can you elaborate on this observation? Why did this frayed appearance happen?

“After 10 days of cells growth in culture, the untreated spheroids appeared frayed and with low nuclear density on the outermost layer, while the ones under TKI treatment (250 and 500 nM) displayed a progressive increase of 3D nuclear density and cell- cell compactness (Fig 3 B).”

• Line 120. Please us subscripts in Na3VO4

Reviewer #4: The manuscript is presented in an intelligible fashion and written in standard English. This study is technically rigorous and meets the scientific standard. Conclusions are drawn appropriately based on the data presented. However, the authors should provide the raw data in the supplementary materials or deposited them to a public repository.

6. PLOS authors have the option to publish the peer review history of their article (what does this mean?). If published, this will include your full peer review and any attached files.

Reviewer #1: No

Reviewer #2: No

Reviewer #3: No

Reviewer #4: No

---

## [Author Response · Author response to Decision Letter 0]

2 May 2021

Reviewer #1 - The manuscript is devoted to an actual issue of modern biomedicine, in particular the development of 3D cell tumor models and their implementation for drug testing. The authors provide some interesting data concerning LoVo spheroids and approaches to evaluate the efficacy of tyrosine kinases inhibitors in tumor spheroids. However, I have some concerns about this manuscript.

Reviewer #1 - Point 1: The novelty of this research should be clarified. The authors use classical approach to form spheroids (low adhesive surface) and commercial kits and equipment, except of fluidic-based device. However, the authors have described this device earlier (https://doi.org/10.3390/mi11050465). Therefore, I suppose that the real novelty of the research is the use of fluidic-based device for antitumor drug testing in tumor spheroids model.

We thank the Reviewer for this insightful comment which helped us to better address and underline the novelty of this approach. Indeed, this study builds upon established technologies, while providing an important step forward in this biomedical approach. As correctly underlined by the reviewer, this study exploits the potential of this new device (previously described by the authors but now addressed in a new applicative clinically relevant scenario), where the new fluidic-based measurements combined with confocal analysis and protein quantification allowed a robust phenotypical characterization of the spheroids structure, offering insights on the spheroids bulk and an accurate measurement of the tumor density. This analysis helps overcome the technical limits of the imaging that hardly penetrates the thickness of 3D structures. This approach could represent a key step forward for the accurate testing of treatment’s potential in a clinically relevant model. 

We better underlined this in the manuscript (starting line 68, now 74):

The innovative use of the new fluidic-based measurements combined with confocal analysis and protein quantification allow to investigate 3D models in detail. A robust morphological characterization supported by biophysical parameters correlation, tested in a challenging and clinically relevant applicative scenario, allowed to unveil a previously unappreciated feature in the tumor spheroids. 

Reviewer #1 - Point 2: The authors use a small ATP-competitive receptor tyrosine kinases inhibitor (TKI), but TKI is not a drug, but a class of drugs (doi:10.2174/138920009788897975). Please clarify which one has been used.

In the revised version of the manuscript, we better clarified the inhibitor used for spheroids treatment. In order to evaluate the ability of our approach to accurately measure spheroids biophysical and morphological parameters, we use LoVo colonspheres, in both control and crizotinib (CZB)-treated conditions. Indeed, this inhibitor has already been proved to impact LoVo spheroids morphology, by increasing cell-cell compactness and reducing spheres size (doi.org/10.1101/2020.10.07.307991). 

We have replaced the TKI nomenclature with crizotinib (CZB) in all parts of the text and in the figures.

We added this sentence at line 70, now 79:

This inhibitor, already been proved to impact LoVo spheroids morphology [26,27],

Reviewer #1 - Point 3: The choice of TKI concentrations is unclear, as well as incubation time (10 days?). If TKI was added to the cells before spheroids formation, it affected on single cell suspension, not on spheroids. Please clarify the protocol.

To this point we would like to explain how this specific issue was detailed in our previous work by an array of both proliferation, apoptosis and survival assays (doi.org/10.1101/2020.10.07.307991). CZB elicits its anti-proliferative/pro-apoptotic effect on LoVo spheroids at the indicated doses (250 and 500 nM). In accordance with the protocol, the inhibitor was added to the medium simultaneously with the seeding, thus we can infer that it acts also on cells ability to aggregate and form colonspheres. 

We added the reference and the following new sentence (line 105, now 123):

The inhibitor was added to the medium simultaneously with the seeding as previously described [27].

Reviewer #1 - Point 4: The use of conventional techniques (MTS, or XTT, or WST-1) could be useful to evaluate the viability of cells in spheroids after TKI treatment. Since these assays have been developed to measure cell viability, they are more sensitive than protein quantification, which is obviously the simple function of cell amount.

We thank the Reviewer for this suggestion. We performed several proliferation assays, by mean of AlamarBlue, on LoVo cells growing in 2D, as monolayer, as previously reported (doi.org/10.1101/2020.10.07.307991). These assays confirmed the efficacy of CZB in vitro. As for the 3D spheroids assay, the proliferation was evaluated by measuring spheroid size and number, which represents a good indicator of cell proliferation, as previously reported [11].

Reviewer #1 - Point 5: How many spheroids was in each well of 24-well plates? I suppose that several dozens, but in this case, Fig. 1 is confusing (one spheroid per well).

In 24-well ultra-low attachment plate are present 554 microcavity per well. 

We revised Fig 1 modifying the representation of the number of spheroids, showing a macro-picture of the cavities and detailing the number of cavity/well.

Reviewer #1 - Point 6: It is better to transfer Fig. 1 from Introduction to Results or even to Graphical abstract since it contains results.

Taking into account the Reviewer’s suggestion we modified Fig 1 removing the results. This new version represents properly the experimental design of the research strategy. 

We modified the figure legend to remove the results (line 75, now 84):

The LoVo cells were cultured in Ultra-Low Attachment Microcavity Plate in presence of 250 and 500 nM of CZB. At 10 days fully mature and organized spheroids were harvested, for the analysis procedures. Live and fixed samples were evaluated with a new fluidic device (W8) for the accurate, simultaneous, and rapid measurement of mass density, weight, and size. These parameters were correlated with protein quantification (NanoOrange Assay) and deep imaging analysis performed on clarified samples.

Reviewer #1 - Point 7: Since the authors made a lot of effort to realize deep imaging, the results (three images of nuclei staining) seem insufficient. I propose to calculate the density of cell nuclei using the total area of spheroids to compare it with mass density data. 

We are grateful for Reviewer’s constructive criticisms and suggestions, which we think were fair and well though. These gave us the opportunity to further strengthen the relevance of our overall findings and improve the level of our work. In order to clearly address this fair request, we added new analysis in Fig 3 D-F: the total number of nuclei per spheroid, the spheroid volume and the cell nuclei density. The data shows a significant increase in the nuclear density confirming the tendency of mass density. We added this information in the “Results and Discussion”, “Figure legends” and in the “Conclusions” sections.

“Results and Discussion” (line 228, now 281)

The quantitative analysis of imaged spheroids, after an accurate automatic segmentation of DAPI signal, indicates a significant decrease in the number of cell nuclei (Fig 3 D) and in spheroid volume (Fig 3 E) of CZB treated samples compared to CTRL. Conversely, under the same treatments, cell nuclei density (the relationship between the number of nuclei and the spheroid volume) displays an opposing result, with a significant increase in the density of the spheroids (Fig 3 F).

“Figure legends” (starting line 213, now 254)

D) Total cell nuclei counting of fixed LoVo spheroids after 10 days of treatment with CZB 250 nM (histogram in black), 500 nM (histogram in gray) and relative control (histogram in white). Statistical analysis was performed using two-tailed unpaired Student’s t-test. *p< 0.05, **p< 0.01, and ***p< 0.001.

E) Volume of fixed LoVo spheroids after 10 days of treatment with CZB 250 nM (histogram in black), 500 nM (histogram in gray) and relative control (histogram in white). Statistical analysis was performed using two-tailed unpaired Student’s t-test. *p< 0.05, **p< 0.01, and ***p< 0.001.

F) Volumetric cell nuclei density of fixed LoVo spheroids, calculated as ratio between the total number of nuclei and the spheroid volume after 10 days of treatment with CZB 250 nM (histogram in black), 500 nM (histogram in gray) and relative control (histogram in white). Statistical analysis was performed using two-tailed unpaired Student’s t-test. *p< 0.05, **p< 0.01, and ***p< 0.001.

“Conclusions” (lines 239 and 240, now 296)

These results were further supported by the protein quantification and the deep imaging analysis that showed protein level decrease in line with a reduced proliferation, a decrease of the number of nuclei and the spheroid volume, but a strong increase in nuclear density and cell-cell compactness, according to mass density data.

Reviewer #1 - Point 7 bis: The phalloidin staining is weak and does not provide any information.

This is an important point highlighted by the Reviewer, and we agree that it was not clear. Fluorescence phalloidin staining provide to demonstrate that there are not significant changes in the actin cytoskeleton organization of the spheroids. However, in the context of a 3D structure like the colonospheres, it may be challenging to point out the actin arrangement of the single cells. The rational of the phalloidin staining was instead to reveal the supporting organization of the whole spheroid. We clarified this observation in the “Results and Discussion” of Fig 3 (line 227, now 280):

Fluorescence phalloidin staining reveal the expression levels of actin cytoskeletal proteins is low and is not affected by CZB treatment.

Reviewer #1 - Point 8: The authors discuss LoVo spheroids as a clinically relevant in vitro model, but in fact, spheroids are in vitro models only, while clinical significance is under discussion.

While spheroids can be acknowledged in the current clinical setting, this could be instead appreciated looking at the field evolution, where 3D models are the focus of increasing attention for providing relevant models to investigate drugs (for example) and their efficacy in the clinical scenario, addressing their mechanism of action but also the responsiveness of specific patients. From a research perspective, this is one of the most important areas holding promise to change the clinical practice. 

We have discussed this part in the manuscript (line 48)

Compared to 2D monolayer cells, spheroids display a 3D morphology, which is closer than 2D monolayer cell system to the tumor bulk in vivo. Spheroids are useful for toxicological studies, as concerning drug penetration and absorption, thus providing insights on drug diffusion [15] and exhibit better immuno-modulatory, proliferation, and activation abilities than 2D cultures [16].

Reviewer #1 - Point 9: The authors discuss the problem of the generation of uniform populations of spheroids, but the manuscript does not provide any new approaches to generate uniform spheroids.

We used one of well know preparation methods to produce uniform size-controlled 3D multicellular tumor spheroids (Ultra-Low Attachment Microcavity Plates) as reported in 10.1016/j.ddtec.2017.03.002. 

We added this information in “Materials and Methods” section (line 103, now 120): [13]

Recent advances in tissue engineering led to the development of a plethora of new techniques to generate 3D spheroid. Despite these methods (such as hanging droplets, pellet cultures, bioreactors) have improved spheroid production, they sometimes require specific laboratory equipment, laborious steps for spheroid manipulation and lead to spheroids with variable morphologies and sizes. In our work, to produce well-controlled size spheroids, an ultra-low attachment microplate, using microcavity technology, was employed. Microcavities allow cells to self-assembly into aggregates with uniform size, depending on microcavity diameters and geometry (DOI: 10.1016/j.ddtec.2017.03.002, DOI: 10.1002/biot.200700228).

We added this information in “Materials and Methods” section (line 103, now 120):

Microcavities allow cells to self-assembly into large numbers of uniform, size-controlled 3D spheroids depending on microcavity diameters and geometry [13,28].

Reviewer #1 - Point 10: Since the authors found that fixation protocols did not alter the spheroids parameters under investigation, these data could be used as supplementary. The fixation protocol (4% PFA) is a standard one in cell biology due to minimal influence on cell properties, so the scientific value of these results are questionable. 

This has been a fair and fundamental criticism that we wanted to address properly.

The aim of live and fixed sample comparison was to investigate whether chemical fixation with paraformaldehyde affects the new parameters obtained from W8 devices. Fixative agents, such as paraformaldehyde, crosslink tissue proteins producing some influence on the MRI parameters (doi.org/10.1002/jor.20767). It remains unknown whether fixation affects fluidic parameter in particular mass density. Moreover, working with fixed samples using standard protocol of fixation avoiding data alteration, allows to perform measurements on stored samples. 

We modified this sentence in the “Results and Discussion” section (line 191, now 227).

These findings confirm that PFA fixation protocol did not altered the parameters under investigation, with respect to MRI parameters [30].

Reviewer #1 - Point 11: In summary, I suppose that the manuscript could not be published in the current version. It contains interesting data concerning the increase of mass density in spheroids after treatment with TKI, but the scientific value is limited due to: 

1) insufficiency in experimental conditions (one drug in two random (?) concentrations in one time point)

See points 2 and 3.

2) absence of conventional techniques to compare, 

We added Cell nuclei density measurement to compare it with mass density data. See point 7. 

3) lack of explanation of mass density increase effect. More experiments are required to demonstrate the importance of mass density as drug efficacy marker, and fluidic-based device as a sensitive approach to measure mass density in comparison to confocal or histology imaging and counting, as well as to conventional assays to evaluate the efficacy of antitumor drugs in 3D cultures (MTS or WST-1). 

See point 4.

The possible mechanism of increase of mass density in spheroids should also be discussed (cell shrinkage due to apoptosis?).

Our data (doi.org/10.1101/2020.10.07.307991) support the hypothesis that the mechanism of action of the CZB treatment is mediated by increased cell-cell adhesion, resulting in enhanced spheroids compactness and diminished metastatic spreading. For these reasons, we speculate that the frayed appearance could be an index of enhanced cells aggressiveness as consequence of the tendency to detach from the main bulk. Of course, these phenotypes will require further characterizations, which beyond the scope of the current manuscript.

We added this sentence (line 244, now 302):

We speculate that the possible mechanism responsible for the increased spheroid mass density may be related to the enhanced cell-cell adhesion, as previously reported [27]. Indeed, we found that CZB treated cells displayed increased e-cadherin production, with consequent improved tight-junction organization. This might lead to an increase in the compactness of the spheroids. Furthermore, we cannot exclude that this phenotype is the consequence of the apoptotic effect registered with CZB inhibition, which might be responsible for a decreased spheroids size and increase in the bulk filling.

Reviewer #2: This research presented an alternative to an important difficulty in the evaluation of spheroids in three-dimensional form, as biophysical aspects mass density, weight, and size of spheroids and morphological aspects by confocal imaging and quantification for evaluation of total protein to correlate with cell mass decrease, reaching its objectives. It is a work of great impact for the area.

We are glad that the Reviewers agree that our results are potentially significant for the area.

Reviewer #2 – Point 1: Material and methods

- about information from lines 103 to 106

“Cells were seeded at the concentration of 1x106 /wells and cultured for 10 days in DMEM (Sigma-Aldrich, St. Louis, MO) with 10% FCS in presence of 250 and 500 nM of TKI (MedChem Express), at 37 °C and 5% CO2. LoVo cells grown in DMEM (Sigma-Aldrich) with 10% FCS were used as control (CTRL). “

Was the spheroid produced simultaneously with the treatment? If not, clarify in the text - MATERIALs AND METHODS lines 103 to 106.

If produced simultaneously with treatment: I suggest evaluating the parameters, in spheroid produced without simultaneous treatment and treated after its formation.

The spheroid was produced for 10 days: It is known that the number of cells and the time of spheroid formation is different for each cell line. As the author mentioned about the difficulty to obtain spheroids of homogeneous size “ the generation of uniform populations of spheroids remains challenging”, I suggest to add in Material and methods about the analyzed characteristics of the spheroid to standardize the 10 days of spheroid formation.

For these assays, cells were treated with the TKI at the seeding time, the text will be clarified accordingly. Concerning the duration of the assay, cells were maintained in culture for the minimum amount of time required to form fully mature and organized spheroids, as shown in doi.org/10.1101/2020.10.07.307991.

We added the reference and the following new sentences. 

(line 105, now 123): The inhibitor was added to the medium simultaneously with the seeding as previously described [27].

(line 106, now 125): At 10 days fully mature and organized spheroids were harvested, for the analysis procedures.

Reviewer #2 – Point 2: RESULTS

About result of fig. 2: Measurements of mass density (A and D), weight (B and E), and diameter (C and F) of 2 methods: live (top panels) and fixed (bottom panels). Since they showed variation in values for each parameter for both methodologies, I suggest statistically evaluating this difference to be able to state "fixation protocols did not alter the parameters under investigation".

We thank the Reviewer for this suggestion. To evaluate the fixation effect in mass density, weight and diameter, we used the "r" Pearson correlation coefficient with the different drug concentrations. Moreover, we estimated the total effect of fixation on W8 output data with one-way ANOVA test.

We added these new sentences:

In "Materials and Methods", "Statistical Analysis" section (starting line 168, now 196):

To evaluate the fixation effect on mass density, weight and diameter, we used the "r" Pearson correlation coefficient (range value from -1 to 1). We analyzed the average value of mass density, weight and diameter for every drug concentration. We estimated the total effect of fixation on W8 output data with one-way ANOVA test as indicated in figure legend.

In "Figure legend" (line 186, now 218):

The "r" Pearson correlation coefficients, able to measure the degree of relationship between the value of mass density, weight and diameter and the different drug concentrations, are indicated for each graph. The F-ratio value obtained from one-way ANOVA test estimating the total effect of fixation on W8 output data, is 0.00027, the result is not significant at p < 0.01.

 

Reviewer #3: I enjoyed very much reading this work. The methods presented here add valuable elements to the toolbox for evaluating 3D microtumors (and microtissues in general). The technology introduced is very clever and elegant. The Figures are very well finished, and the manuscript flows very nice.

We thank the Reviewer #3 for Her/His enthusiastic appreciation of our work. Concerning the reviewer’s remarks, we proceed as follow:

Reviewer #3 – point 1: The fluidic device is equipped with a brightfield setup. Could you add a panel in figure 1 or a supplementary video showing how the spheroids look like when passing through the channel? 

We added a new panel in figure 1 showing the field of view captured by the detection unit of a representative control spheroid passing through the fluidic core-chip.

Moreover, we added a Supporting Information Video 1 displaying the effect of the forces on a falling spheroid within the analysis channel during the measurement process. We added the link in line 86, now 96.

(see Supporting Information Video 1)

Video 1 Legend (starting line 96, now 108)

Supporting Information Video 1 

Representative video displaying the different phases of the W8 measurement system analysis: spheroid selection, stabilizing and centering phases, measurements process (2 repetitions), and the final sorting. The analysis channel is shown at low magnification (LQ camera, on the left) and at higher magnification (HQ camera, on the right). The frame rate speeded up about 3 times.

Reviewer #3 – point 2: The results of protein concentrations with NanoOrange assay is repeated in Figure 1 and 3. Instead, present the results only in Figure 3, and use and schematic representation of the assay in Figure 1 (a microplate reader?). 

Taking into account the Reviewer’s suggestion we modified Fig 1 removing the results present. We plotted the standard curve as graphical representation of NanoOrange assay. This new version represents properly the experimental design of the research strategy. 

Moreover, we modified the Figure 1 legend (line 75, now 84):

The LoVo cells were cultured in Ultra-Low Attachment Microcavity Plate in presence of 250 and 500 nM of CZB. At 10 days fully mature and organized spheroids were harvested, for the analysis procedures. Live and fixed samples were evaluated with a new fluidic device (W8) for the accurate, simultaneous, and rapid measurement of mass density, weight, and size. These parameters were correlated with protein quantification (NanoOrange Assay) and deep imaging analysis performed on clarified samples.

Reviewer #3 – point 3: Figure 3. Could you present the nuclei count in histograms with error bars? Are these nuclei the total number in the spheroid (volumetrically)? 

We are grateful for Reviewer’s constructive suggestion. In order to clearly address this request, we presented the total number of nuclei per spheroid in Fig 3 D with the relative statistical analysis. We added this information in the “Results and Discussion” and in the “Figure legends” sections.

“Results and Discussion” (line 228, now 282)

The quantitative analysis of imaged spheroids, after an accurate automatic segmentation of DAPI signal, indicates a significant decrease in the total number of cell nuclei (Fig 3 D)

“Figure legends” (starting line 213, now 254)

D) Total cell nuclei counting of fixed LoVo spheroids after 10 days of treatment with CZB 250 nM (histogram in black), 500 nM (histogram in gray) and relative control (histogram in white). Statistical analysis was performed using two-tailed unpaired Student’s t-test. *p< 0.05, **p< 0.01, and ***p< 0.001.

Reviewer #3 – point 4: Could you add an additional panel in Figure 3 or a supplementary video showing the volumetric nature of the sample? 

We added a new panel in figure 3 B showing 3D z-depth rendering of the spheroids. We added this information in the “Figure legends” section.

“Figure legends” (line 221, now 248)

The lower panel provides volume view of DAPI signal in 3D z-depth confocal rendering reported as a scale of colors (palette on the right).

Reviewer #3 – point 5: Please add titles to the axis (Y axis: Protein concentration (�g/ml); X axis: Sample) 

We added the titles “Protein Concentration (�g/ml)” to the Y axis and the name of the samples to the X axis in Fig 3 A.

Reviewer #3 – point 6: Can you elaborate on this observation? Why did this frayed appearance happen?

“After 10 days of cells growth in culture, the untreated spheroids appeared frayed and with low nuclear density on the outermost layer, while the ones under TKI treatment (250 and 500 nM) displayed a progressive increase of 3D nuclear density and cell- cell compactness (Fig 3 B).” 

Our data (doi.org/10.1101/2020.10.07.307991) support the hypothesis that the mechanism of action of the TKI treatment is mediated by increased cell-cell adhesion, resulting in enhanced spheroids compactness and diminished metastatic spreading. For these reasons, we speculate that the frayed appearance could be an index of enhanced cells aggressiveness as consequence of the tendency to detach from the main bulk. Of course, these phenotypes will require further characterizations, which beyond the scope of the current manuscript.

We added this sentence (line 244, now 302):

We speculate that the possible mechanism responsible for the increased spheroid mass density may be related to the enhanced cell-cell adhesion, as previously reported [27]. Indeed, we found that CZB treated cells displayed increased e-cadherin production, with consequent improved tight-junction organization. This might lead to an increase in the compactness of the spheroids. Furthermore, we cannot exclude that this phenotype is the consequence of the apoptotic effect registered with CZB inhibition, which might be responsible for a decreased spheroids size and increase in the bulk filling.

Reviewer #3 – point 7: Line 120. Please us subscripts in Na3VO4 

We corrected in Na3VO4 

 

Reviewer #4: The manuscript is presented in an intelligible fashion and written in standard English. This study is technically rigorous and meets the scientific standard. Conclusions are drawn appropriately based on the data presented. However, the authors should provide the raw data in the supplementary materials or deposited them to a public repository.

We are glad that the Reviewers agree that our conclusions are drawn appropriately based on the data presented. Concerning the reviewer’s remark, we added a new section of “Supporting Information” in “Materials and Methods” containing the raw data of sample measurements and repetitions performed by W8 fluidic device.

We added this sentence in line 91, now 102:

Raw data can be found in Supporting Information Dataset 1. 

Reviewer #5 Weiren Huang

The manuscript involves a 3D culture system to study the biophysical and morphological parameters in LoVo spheroids treated with TKI. The authors used a new fluidic-based measurement, combined with confocal imaging and protein quantification methods, to show that they can provide robust phenotypical characterization of the spheroids structure, offering insights on the spheroids bulk and an accurate measurement of the tumor density. They also represented that this analysis helps overcome the technical limits of the imaging that it is difficult to penetrate the thickness of 3D structures. More importantly, they demonstrated that TKI treatment has an impact on mass density, which represents a key marker of cancer cell treatment. Overall, the accurate characterization of in vitro spheroid models is of great significance for cancer research and clinical treatment. The experiments described here are well designed but there are some concerns that require additional analysis and/or discussion.

We are glad that the Reviewers agree that our results are of great significance for cancer research and clinical treatment. Concerning the reviewer’s remarks, we will proceed as follow:

Reviewer #5 – point 1: I wonder if the W8 fluidic system described in this study is applicable to some other cell types? If yes, please point out the relevant evidence. If no, please explain the reason. 

We added a new reference regarding one our recently published work on four different adenocarcinoma cell lines and NK cells. 

We also modified sentence on line 61, now 65:

To address this issue, we recently developed a new device (W8, CellDynamics iSRL) and relative analytical method, for the accurate, simultaneous, and rapid measurement of mass density, weight, and size of spheroids generated from different cell types [24,25]. In general, the device is able to measure all types of spheroids derived from cancer and primary cells which meet the operative range between 50 and 500 µm in diameter.

Reviewer #5 – point 2: Materials and Methods: 

a. Line 115, the reference “Cristaldi et al.” should be cited with the number [22] in the text. 

We added the reference number.

b. Line 121, a brief description of the NanoOrange methods should be provided. 

We added this new sentence (line 122, now 143):

Briefly, samples were diluted in the NanoOrange working solution and incubated at 90°C for 10 minutes, protected from light. Successively, samples were cooled at room temperature for at least 20 minutes and transferred in a 96 wells plate. Fluorescence was read at 470ex–570em nm wavelength, using a Spectra Max Gemini plate fluorometer (Molecular Devices, Sunnyvale, CA). Protein concentration was determined using the reference standard curve (R2, coefficient of determination was reported in Fig 1).

Reviewer #5 – point 3: Results and Discussion

a. Bright field images of spheroid cultures in the presence or absence of TKI should be provided, so that we can observe their morphological characteristics during culture. 

We added a new panel in figure 3 B showing bright field imaging at low magnification. We added this information in the “Figure legends” section.

“Figure legends” (line 211, now 246)

The upper panel shows bright field imaging from control and treated spheroids at low magnification.

Moreover, we performed optical density measurements (Fig 3 C) of brightfield images. We added this information in the “Figure legends” and “Results and Discussion” sections.

“Results and Discussion” (starting line 219, now 270)

Bright field images from clarified samples (Fig 3 B upper panel) show a decrease in size and an increase in optical density (as measured in Fig 3 C), indicating a progressive growth in cell-cell compactness.

“Figure legends” (starting line 213, now 250)

C) Optical density of fixed LoVo spheroids, calculated as ratio between the mode of absorbance (subtracted to the background) and the spheroid area after 10 days of treatment with CZB 250 nM (histogram in black), 500 nM (histogram in gray) and relative control (histogram in white). Statistical analysis was performed using two-tailed unpaired Student’s t-test. *p< 0.05, **p< 0.01, and ***p< 0.001.

b. Line 177-178 “This is in line with an antiproliferative activity of the inhibitor”, please add a citation source. 

We added the reference [27].

Reviewer #5 – point 4: Figure 3B: 

a. All the fluorescent images provided only show single spheroid. Please provide a more comprehensive view at lower magnification to capture more spheroids. 

We added a new panel in figure 3 B showing bright field imaging at low magnification. We added this information in the “Figure legends” section.

“Figure legends” (line 221, now 246)

The upper panel shows bright field imaging from control and treated spheroids at low magnification.

b. The authors do not show how many spheroids were analyzed in each group, and how many times were these experiments repeated? In addition to showing representative fluorescent images, the authors need to present these results with statistical analysis. 

We added how many spheroids were analyzed in each group, and how many times were these experiments repeated in line 158, now 185:

A minimum of 10 single spheroids were analyzed for every tested condition and performed in triplicate.

Moreover, we added a new statistical analysis in Fig 3 C-E: the total number of nuclei per spheroid, the spheroid volume and the cell nuclei density. 

We added this information in the “Results and Discussion”, “Figure legends” and in the “Conclusions” sections.

“Results and Discussion” (line 228, now 282)

The quantitative analysis of imaged spheroids, after an accurate automatic segmentation of DAPI signal, indicates a significant decrease in the total number of cell nuclei (Fig 3 D) and in spheroid volume (Fig 3 E) of CZB treated samples compared to CTRL. Conversely, under the same treatments, cell nuclei density (the relationship between the number of nuclei and the spheroid volume) displays an opposing result, with a significant increase in the density of the spheroids (Fig. 3 F).

“Figure legends” (starting line 213, now 254)

D) Total cell nuclei counting of fixed LoVo spheroids after 10 days of treatment with CZB 250 nM (histogram in black), 500 nM (histogram in gray) and relative control (histogram in white). Statistical analysis was performed using two-tailed unpaired Student’s t-test. *p< 0.05, **p< 0.01, and ***p< 0.001.

E) Volume of fixed LoVo spheroids after 10 days of treatment with CZB 250 nM (histogram in black), 500 nM (histogram in gray) and relative control (histogram in white). Statistical analysis was performed using two-tailed unpaired Student’s t-test. *p< 0.05, **p< 0.01, and ***p< 0.001.

F) Volumetric cell nuclei density of fixed LoVo spheroids, calculated as ratio between the total number of nuclei and the spheroid volume after 10 days of treatment with CZB 250 nM (histogram in black), 500 nM (histogram in gray) and relative control (histogram in white). Statistical analysis was performed using two-tailed unpaired Student’s t-test. *p< 0.05, **p< 0.01, and ***p< 0.001.

“Conclusions” (lines 239 and 240, now 296)

These results were further supported by the protein quantification and the deep imaging analysis that showed protein level decrease in line with a reduced proliferation, a decrease of the number of nuclei and the spheroid volume, but a strong increase in nuclear density and cell-cell compactness, according to mass density data.

Reviewer #5 – point 5: Patient-derived organoid models are thought to be the next generation in vitro models that have been shown to closely recapitulate patient's disease. Unlike cell lines, organoids are both genetically and phenotypically more stable and recapitulate the main features of patient’s tumor tissues. In terms of preclinical studies, organoid may be more suitable for drug screening and tumor biology research than spheroids derived from self-organization of cancer cell lines. Please discuss this point. Thank you. 

We thank the reviewer for this important insight in the field which is definitely worth underlining. We actually decided to include this concept right at the end, to underline this issue, while showing the importance to confirm the study findings also in models of increasing complexity to improve personalized medicine. In detail, we added this new sentence in line 244, now 309:

All these and further key characteristics in tumor development and response to treatment can be better investigated in 3D models. In this light, the results of the newly proposed methodology could serve as a stepping stone to further improve the 3D model field. The use of patient-derived 3D tumor models can be considered the next generation of in vitro models that can closely recapitulate patient’s disease and can be used to screen drugs, to optimized the correct dosage, as well as to develop new personalized therapeutic strategies [32]. However, the disadvantages of 3D models include difficulties evaluating biophysical parameters and obtaining high-resolution imaging analysis due to hardly light penetration in the thickness of deep structures. While this study provided the proof of concept in spheroids to support this methodology, this promising approach should be also applied to more complex patient-derived 3D systems in the field of personalized medicine. Thus, future studies should aim at confirming the positive findings obtained in spheroids also in organoids and 3D models of increasing complexity, where the adoption of such precise measurement of the tumor characteristics could represent a key step forward for the accurate testing of treatment’s potential in a clinically relevant model.

 

List of added references

[15] G.J. LaBonia, S.Y. Lockwood, A.A. Heller, D.M. Spence, A.B. Hummon, Drug penetration and metabolism in 3D cell cultures treated in a 3D printed fluidic device: assessment of irinotecan via MALDI imaging mass spectrometry, PROTEOMICS. 16 (2016) 1814–1821. https://doi.org/10.1002/pmic.201500524.

[16] K. Ramgolam, J. Lauriol, C. Lalou, L. Lauden, L. Michel, P. de la Grange, A.-M. Khatib, F. Aoudjit, D. Charron, C. Alcaide-Loridan, R. Al-Daccak, Melanoma Spheroids Grown Under Neural Crest Cell Conditions Are Highly Plastic Migratory/Invasive Tumor Cells Endowed with Immunomodulator Function, PLOS ONE. 6 (2011) e18784. https://doi.org/10.1371/journal.pone.0018784.

[25] A. Sargenti, F. Musmeci, F. Bacchi, C. Delprete, D.A. Cristaldi, F. Cannas, S. Bonetti, S. Pasqua, D. Gazzola, D. Costa, F. Villa, M.R. Zocchi, A. Poggi, Physical Characterization of Colorectal Cancer Spheroids and Evaluation of NK Cell Infiltration Through a Flow-Based Analysis, Front. Immunol. 11 (2020). https://doi.org/10.3389/fimmu.2020.564887.

[28] R.-Z. Lin, R.-Z. Lin, H.-Y. Chang, Recent advances in three-dimensional multicellular spheroid culture for biomedical research, Biotechnol J. 3 (2008) 1172–1184. https://doi.org/10.1002/biot.200700228

[30] A. Dugar, M.L. Farley, A.L. Wang, M.B. Goldring, S.R. Goldring, B.H. Swaim, B.E. Bierbaum, D. Burstein, M.L. Gray, The effect of paraformaldehyde fixation on the Delayed Gadolinium-Enhanced MRI of Cartilage (dGEMRIC) measurement, Journal of Orthopaedic Research. 27 (2009) 536–539. https://doi.org/10.1002/jor.2076

[32] Z. Gilazieva, A. Ponomarev, C. Rutland, A. Rizvanov, V. Solovyeva, Promising Applications of Tumor Spheroids and Organoids for Personalized Medicine, Cancers. 12 (2020) 2727. https://doi.org/10.3390/cancers12102727.

---

## [Decision Letter · Decision Letter 1]

25 May 2021

A new method for the study of biophysical and morphological parameters in 3D cell cultures: Evaluation in LoVo spheroids treated with crizotinib

PONE-D-21-04842R1

Dear Dr. Santi,

We’re pleased to inform you that your manuscript has been judged scientifically suitable for publication and will be formally accepted for publication once it meets all outstanding technical requirements.

Kind regards,

Irina V. Balalaeva, PhD

Academic Editor

PLOS ONE

Additional Editor Comments (optional):

Reviewers' comments:

Reviewer's Responses to Questions

**Comments to the Author**

1. If the authors have adequately addressed your comments raised in a previous round of review and you feel that this manuscript is now acceptable for publication, you may indicate that here to bypass the “Comments to the Author” section, enter your conflict of interest statement in the “Confidential to Editor” section, and submit your "Accept" recommendation.

Reviewer #1: All comments have been addressed

Reviewer #2: All comments have been addressed

Reviewer #3: All comments have been addressed

2. Is the manuscript technically sound, and do the data support the conclusions?

Reviewer #1: Yes

Reviewer #2: Yes

Reviewer #3: Yes

3. Has the statistical analysis been performed appropriately and rigorously? 

Reviewer #1: Yes

Reviewer #2: Yes

Reviewer #3: Yes

4. Have the authors made all data underlying the findings in their manuscript fully available?

Reviewer #1: Yes

Reviewer #2: Yes

Reviewer #3: Yes

5. Is the manuscript presented in an intelligible fashion and written in standard English?

Reviewer #1: Yes

Reviewer #2: Yes

Reviewer #3: Yes

6. Review Comments to the Author

Reviewer #1: (No Response)

Reviewer #2: The authors have adjusted the previous comments. They have clarified methodological issues that were previously doubtful. The statistical adjustments in the text were important for the conclusion. This research presented an alternative to an importante difficulty in the evaluation of spheroids in three-dimensional form, as biophysical aspects mass density, weight, and size of spheroids and morphological aspects for evaluating molecular aspects of diseases, mimicking what happens in vivo. In this way represents a great advance for the use of this methodology - spheroids in three-dimensional form.

Reviewer #3: The authors have diligently addressed all the comments of this reviewer to satisfaction.

My reccomendation is "Accept" the manuscript in its current form.

7. PLOS authors have the option to publish the peer review history of their article (what does this mean?). If published, this will include your full peer review and any attached files.

Reviewer #1: No

Reviewer #2: No

Reviewer #3: **Yes: **Grissel Trujillo de Santiago

---

## [Editor Report · Acceptance letter]

31 May 2021

PONE-D-21-04842R1 

A new method for the study of biophysical and morphological parameters in 3D cell cultures: Evaluation in LoVo spheroids treated with crizotinib 

Dear Dr. Santi:

I'm pleased to inform you that your manuscript has been deemed suitable for publication in PLOS ONE. Congratulations! Your manuscript is now with our production department. 

Kind regards, 

on behalf of

Dr. Irina V. Balalaeva 

Academic Editor

PLOS ONE